# Improved prediction of protein-protein interactions using AlphaFold2

Patrick Bryant [1,2,3✉], Gabriele Pozzati [1,2,3] & Arne Elofsson [1,2✉]

Predicting the structure of interacting protein chains is a fundamental step towards understanding protein function. Unfortunately, no computational method can produce accurate structures of protein complexes. AlphaFold2, has shown unprecedented levels of accuracy in modelling single chain protein structures. Here, we apply AlphaFold2 for the prediction of heterodimeric protein complexes. We find that the AlphaFold2 protocol together with optimised multiple sequence alignments, generate models with acceptable quality (DockQ ≥ 0.23) for 63% of the dimers. From the predicted interfaces we create a simple function to predict the DockQ score which distinguishes acceptable from incorrect models as well as interacting from non-interacting proteins with state-of-art accuracy. We find that, using the predicted DockQ scores, we can identify 51% of all interacting pairs at 1% FPR.

[1] Science for Life Laboratory, 172 21 Solna, Sweden. [2] Department of Biochemistry and Biophysics, Stockholm University, 106 91 Stockholm, Sweden. [3]These authors contributed equally: Patrick Bryant, Gabriele Pozzati. ✉email: patrick.bryant@scilifelab.se; arne@bioinfo.se

Protein–protein interactions are central mediators in biological processes. Most interactions are governed by the three-dimensional arrangement and the dynamics of the interacting proteins[1]. Such interactions vary from being permanent to transient[2,3]. Some protein–protein interactions are specific for a pair of proteins, while some proteins are promiscuous and interact with many partners. This complexity of interactions is a challenge both for experimental and computational methods.

Often, studies of protein–protein interactions can be divided into two categories, the identification of what proteins interact and the identification of how they interact. Although these problems are distinguished, some methods have been applied to both problems[4,5]. Protein docking methodologies refer to how proteins interact and can be divided into two categories considering proteins as rigid bodies; those based on an exhaustive search of the docking space[6] and those based on alignments (both sequence and structure) to structural templates[7]. Exhaustive approaches rely on generating all possible configurations between protein structures or models of the monomers[8,9] and selecting the correct docking through a scoring function, while template-based docking only needs suitable templates to identify a few likely candidates. However, flexibility has often to be considered in protein docking to account for interaction-induced structural rearrangements[10,11]. Therefore, flexibility limits the accuracy achievable by rigid-body docking[12], and flexible docking is traditionally too slow for large-scale applications. A possible compromise is represented by semi-flexible docking approaches[13] that are more computationally feasible and can consider flexibility to some degree during docking.

Regardless of different strategies, docking remains a challenging problem. In the CASP13-CAPRI experiments, human group predictors achieved up to 50% success rate (SR) for top-ranked docking solutions[14]. Alternatively, a recent benchmark study[8] reports SRs of different web-servers reaching up to 16% on the well-known Benchmark 5 dataset[15].

Recently, in the CASP14 experiment, AlphaFold2 (AF2) reached an unprecedented performance level in structure prediction of single-chain proteins[16]. Thanks to an advanced deep learning model that efficiently utilises evolutionary and structural information, this method consistently outperformed all competitors, reaching an average GDT_TS score of 90[16]. Recently, RoseTTAFold was developed, trying to implement similar principles[17]. Since then, other end-to-end structure predictors have emerged using different principles such as fast multiple sequence alignment (MSA) processing in DMPFold2[18] and language model representations[19].

As an alternative to other docking methods, it is possible to utilise co-evolution to predict the interaction between two protein chains. Initially, direct coupling analysis (DCA) was used to predict the interaction of bacterial two-component signalling proteins[20,21]. Later, these methods were improved using machine learning[22].

In a Fold and Dock approach, two proteins are folded and docked simultaneously. We recently developed a Fold and Dock pipeline using another distance prediction method focused on protein folding (trRosetta[23]). In this pipeline, the interaction between two chains from a heterodimeric protein complex and their structures were predicted using distance and angle constraints from trRosetta[24,25]. This study demonstrated that a pipeline focused on intra-chain structural feature extraction can be successfully extended to derive inter-chain features as well. Still, only 7% of the tested proteins were successfully folded and docked.

In that study, we found that generating the optimal MSA is crucial for obtaining accurate Fold and Dock solutions, but this is not always trivial due to the necessity to identify the exact set of interacting protein pairs[26]. Given the existence of multiple paralogs for most eukaryotic proteins, this is difficult. We also found that this process requires an optimal MSA depth to optimise inter-chain information extraction. Too deep MSAs might contain false positives (i.e. protein pairs that interact differently), resulting in noise masking the sought after co-evolutionary signal, while too shallow alignments do not provide sufficient co-evolutionary signals.

In this work, we systematically apply the AF2 pipeline on two different datasets to Fold and Dock protein–protein pairs simultaneously. We explore the docking success using the AF2 pipeline in combination with different input MSAs, in order to study the relationship between the output model quality and these inputs. We also find that, by scoring multiple models of the same protein–protein interaction with a predicted DockQ score (pDockQ), we can distinguish with high confidence acceptable (DockQ ≥ 0.23) from incorrect models. The modelling success is higher for bacterial protein pairs, pairs with large interaction areas consisting of helices or sheets, and many homologous sequences. We also test the possibility to distinguish interacting from non-interacting proteins and find that, using pDockQ, we can separate truly interacting from non-interacting proteins with consistent accuracy. We find that the results in terms of successful docking using AF2 are superior to other docking methods. AF2 clearly outperforms a recent state-of-the-art method[27] and our protocol performs quite close to (63% vs 72%) the recently developed AF-multimer[28], which was developed using the same data as the test set here, making a direct comparison difficult.

## Results and discussion

**Identifying the best AlphaFold2 model.** The SR, i.e., the percentage of acceptable models (DockQ > 0.23), is used to measure AF2 performance over the development set (216 proteins) using the different MSAs. The best performance is 33.3% for the AF2 MSAs and 39.4% for the AF2+ paired MSAs (Table 1). It is thereby evident that combining both paired and AF2 MSAs is superior to using them separately. The average performance of the AF2 and the paired MSAs is similar, but for individual protein pairs, frequently one of the two MSAs is superior to the other, as seen from that the Pearson correlation coefficient for the DockQ scores between AF2 vs paired MSAs is 0.54 (Supplementary Table 1). Therefore, combining AF2 and paired MSAs improves the results.

Next, we compared the default AF2 model (model_1) with the fine-tuned versions of (model_1_ptm). Surprisingly, the original AF2 model_1 outperforms AF2 model_1_ptm in most cases (Table 1). Further, the difference between 10 recycles-one ensemble and three recycles-eight ensembles is minor across all MSAs and AF2 models. Therefore, the input information and the AF2 model appear to impact the outcome the most.

---

**Table 1 Success rate of different modelling setups.**

**Neural network configuration**

| NN model | model_1 | model_1 | model_1_ptm | model_1_ptm |
|---|---|---|---|---|
| Recycles | 10 | 3 | 10 | 3 |
| Ensembles | 1 | 8 | 1 | 8 |
| Setup short name | m1-10-1 | m1-3-8 | mp-10-1 | mp-3-8 |
| Paired MSA | **28.7** | 28.2 | **28.7** | 27.8 |
| AF2 MSA | 31.5 | **33.3** | 26.4 | 23.6 |
| AF2+Paired MSA | **39.4** | 38.4 | 32.4 | 31.0 |

Results of AF2 run on the development set ($n = 216$) using different MSAs and neural network configurations. Row labels in bold indicate AF2 setup features. Every column in the table refers to an overall setup and every corresponding value refers to a run of the described setup with a different input MSA. Values represent the percentage of acceptable models (DockQ ≥ 0.23) overall the development set. The highest success rates for each MSA type are highlighted in bold.

---

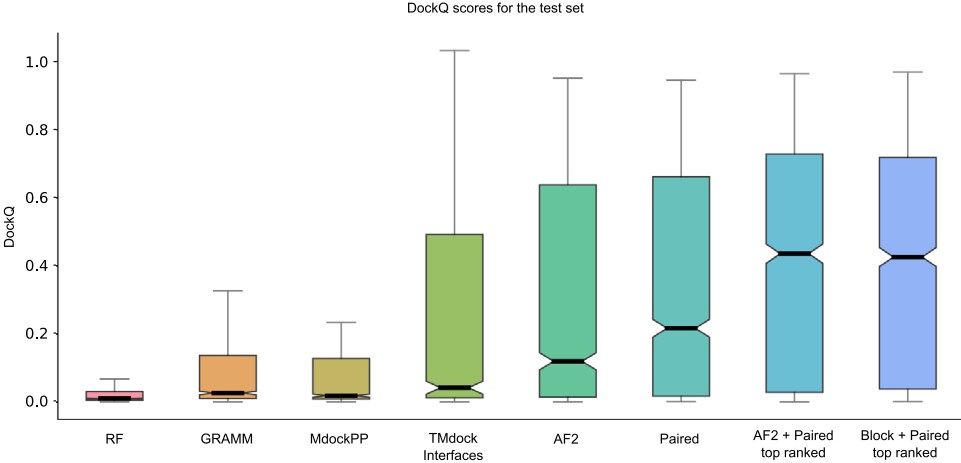

**Fig. 1 DockQ scores for the test set (*n* = 1481 for all but RF, *n* = 1455).** Distribution of DockQ scores as boxplots for different modelling strategies on the test set. Boxes encompass data quartiles, horizontal lines mark the medians and upper and lower whiskers indicate respectively maximum and minimum values for each distribution. All AF2 models have been run with the same neural network configuration (m1-10-1). Outlier points are not displayed here. AF2, refers to running AF2 using the default AF2 MSAs, "Paired" refers to using MSAs paired using information about species and "Block" refers to using block diagonalization MSAs.

**Test set performance**. The best model and configuration for AF2 (m1-10-1) was used for further studies on the test set. The best outcome using this modelling strategy results in an SR of 57.8% (856 out of 1481 correctly modelled complexes) for the AF2 + paired MSAs compared with 45.0% using the AF2 MSAs alone (Fig. 1, Table 2). The results using the block diagonalization+paired MSAs are almost identical (SR = 58.4%, median = 0.363). Further, running five initialisations with random seeds and ranking the models using the predicted DockQ score (pDockQ, Fig. 2c), increases the SR to 61.7% and 62.7% for the AF2 + paired and block diagonalization +paired MSAs, respectively (model variation and ranking, Fig. 2). Using the combination of AF2 and paired MSAs increases performance, suggesting that AF2 gains both from larger and paired MSAs, although it often can manage with less information.

What is most striking is that AF2 outperforms all other tested docking methods by a large margin (Fig. 1, Table 2). RF is better than AF2 only for 14 pairs in the test set, while GRAMM and template-based docking (TMdock interface) outperform AF2 for 188 and 225 pairs, respectively. The best performing method in the CASP14-CAPRI experiment[29], MDockPP[30], achieves a SR of only 24.2%. The reason GRAMM, TMdock and MDockPP reach this level of performance is likely due to the use of the bound form of the proteins, resulting in very high shape complementarity and therefore having the "answer" provided in a way.

The recently developed AF-multimer[28] has the best performance (SR = 72.2%, median = 0.560, Table 2). This method was trained using the same data as the test set, which makes a direct comparison difficult. Regardless, we do believe it is likely that using AF-multimer, the performance would increase over the results of our pipeline, but it is possible the difference is less than the observed 9%.

**Distinguishing acceptable from incorrect models**. It is not only essential to obtain improved predictions, but also to be able to discriminate between acceptable and non-acceptable ones. We measure the separation between correct (DockQ ≥ 0.23) and incorrect models provided by several metrics using a receiver operating characteristic (ROC) curve. Different criteria were examined over the test set, including (i) the number of unique interacting residues (Cβ atoms from different chains within 8 Å from each other) in the interface, (ii) the total number of interactions between Cβ atoms in the interface, (iii) the average

**Table 2 Success rate and median DockQ scores for the test set using different methods and model configurations.**

| Method | Success rate (%) | Median DockQ |
|---|---|---|
| RoseTTAfold | 9.6 | 0.011 |
| GRAMM | 21.4 | 0.027 |
| MDockPP | 24.2 | 0.019 |
| TMdock | 33.6 | 0.040 |
| TMdock interfaces | 35.1 | 0.042 |
| AlphaFold2 | 45.0 | 0.120 |
| Paired | 49.6 | 0.217 |
| AlphaFold2 + Paired | 57.8 | 0.382 |
| Block + Paired | 58.4 | 0.363 |
| AlphaFold2 + Paired top ranked | 61.7 | 0.436 |
| Block + Paired top ranked | 62.7 | 0.426 |
| AF-multimer | 72.2% | 0.560 |

"Block" refers to block diagonalization MSAs. Results for different docking methods in terms of success rate and median DockQ scores on the test set. The number of complexes is *n* = 1481 for all methods except for RoseTTAfold (*n* = 1455) and AF-multimer (*n* = 1458).

plDDT for the interface, (iv) the lowest plDDT of each single-chain average, and (v) the average plDDT over the whole protein heterodimer (Fig. 2a). Three criteria result in very similar areas under the curve (AUC) measures. The total number of interactions between Cβs and the number of residues in the interface can separate the correct/incorrect models with an AUC of 0.92 and 0.91 respectively, while the average interface plDDT results in an AUC of 0.88. However, plDDT results in higher TPRs at lower FPRs; therefore, we multiply the plDDT with the logarithm of the interface contacts resulting in an AUC of 0.95.

Interestingly, the average plDDT of the entire complex only results in an AUC of 0.66, suggesting that both single chains in a complex are often predicted very well, while their relative orientation may still be incorrect.

Figure 2b shows that increasing both the number of interface contacts and the average interface plDDT results in higher DockQ scores for the test set. Using the combination of plDDT with the logarithm of the interface contacts, we, therefore, fit a simple sigmoidal function to the DockQ scores (Fig. 2c), see methods. This enables the prediction of the DockQ scores

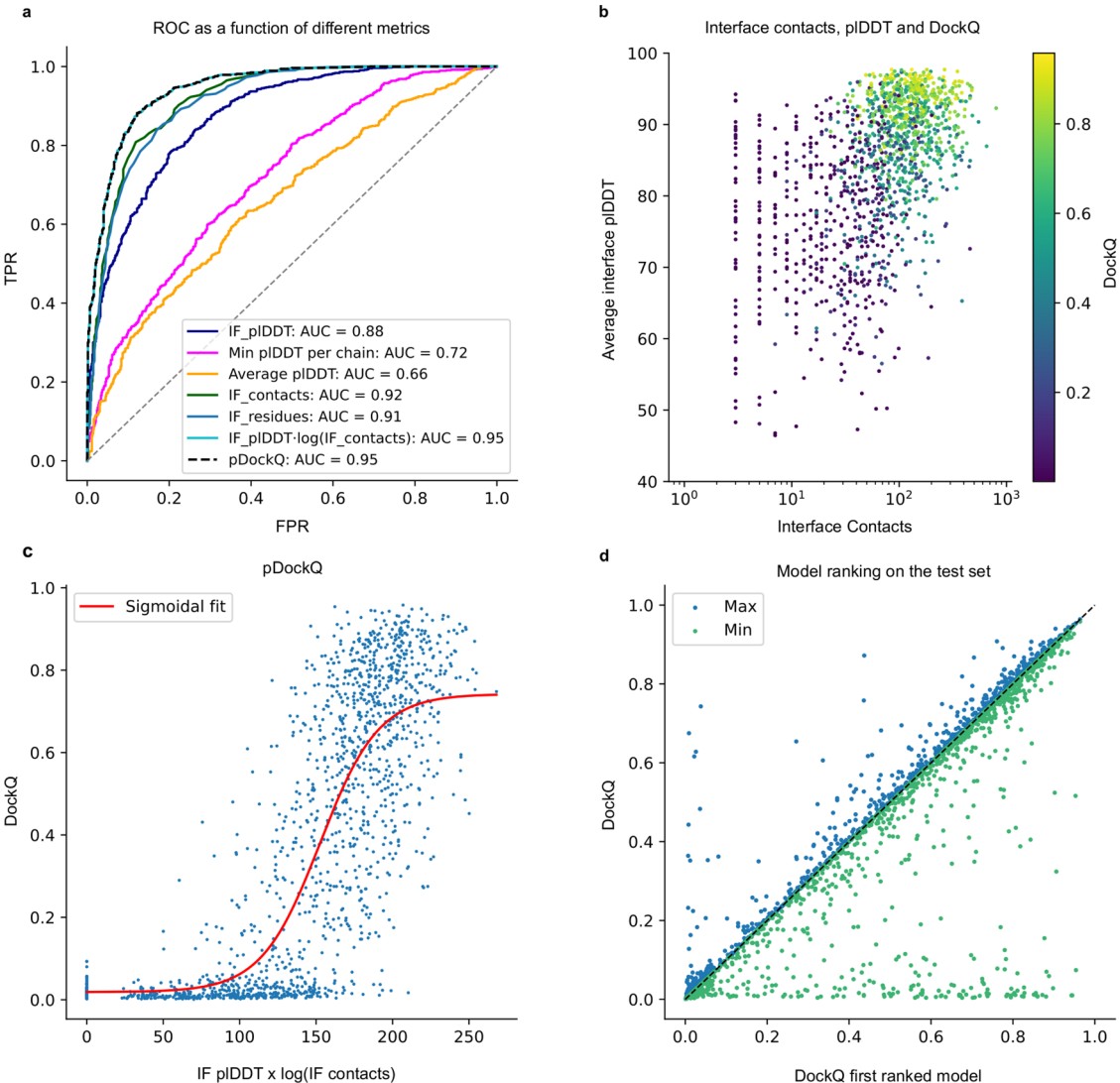

**Fig. 2 Model quality metrics and multiple model ranking. a** ROC curve as a function of different metrics for the test dataset (*n* = 1481, first run). Cβs within 8 Å from each other from different chains are used to define the interface. IF_plDDT is the average plDDT of interface residues, min plDDT per chain is the minimum average plDDT of both chains, average plDDT is the average of the entire complex and IF_contacts and IF_residues are the number of interface residues and contacts respectively. pDockQ is a sigmoidal fit to the combined metric IF_plDDT·log(IF_contacts) fitted to predict DockQ as the target score, see C. **b** Average interface plDDT vs the logarithm of the interface contacts coloured by DockQ score on the test set (*n* = 1481). Increasing both the number of interface contacts and average interface plDDT results in higher DockQ scores. **c** Using the combined metric IF_plDDT·log(IF_contacts), we fit a sigmoidal curve towards the DockQ scores on the test set (*n* = 1481), enabling predicting the DockQ score in a continuous manner (pDockQ). The average error overall is 0.14 DockQ score. **d** Impact of different initialisations on the modelling outcome in terms of DockQ score on the test dataset (*n* = 1481). The maximal and minimal scores are plotted against the top-ranked models using the pDockQ scores for the AF2 + paired MSAs, m1-10-1.

(pDockQ) in a continuous manner with an overall average error of 0.11 on the test set. The AUC using pDockQ as a separator is identical to the combination of plDDT with the logarithm of the interface contacts, 0.95 (Fig. 2a).

**Model variation and ranking for the test set**. Five models are generated using the best strategy (m1-10-1 with AF2 + paired MSAs) with different initialisation (random seeds). The average SR (57.2% ± 0.0%) is similar for all five runs. However, the average deviation for individual models is DockQ = 0.08 when comparing the best and worst models for a target (Fig. 2d), i.e., there is some randomness to the success for an individual pair. If the maximal DockQ score across all models is used, the SR would be 62.9%. Although this is unachievable, ranking the models using the pDockQ score results in an SR of 61.7%. The AUC using the same metric for the ranked test set is 0.93, which means

that 31% of all models are acceptable at an error rate of 1% and 54% at an error rate of 10% (Supplementary Table 2).

**Bacterial complexes are predicted more accurately**. In the test set, about 60% of the complexes can be modelled correctly. We try to identify what distinguishes the successful and unsuccessful cases by analysing different subsets of the test set. First, we divide the proteins by taxa, next by interface characteristics and finally by examining the alignments.

The SRs for each kingdom is; Eukarya 61%, Bacteria 73.7%, Archaea 84.5%, and Virus 60% (Supplementary Fig. 1b). Further, the SRs for *Saccharomyces cerevisiae* is better than for *Homo sapiens* (66% vs 58%, Fig. 3d). The higher performance in prokaryotes is consistent with previous observations regarding the availability of evolutionary information in prokaryotes compared to Eukarya[27] (Supplementary Fig. 1). The higher

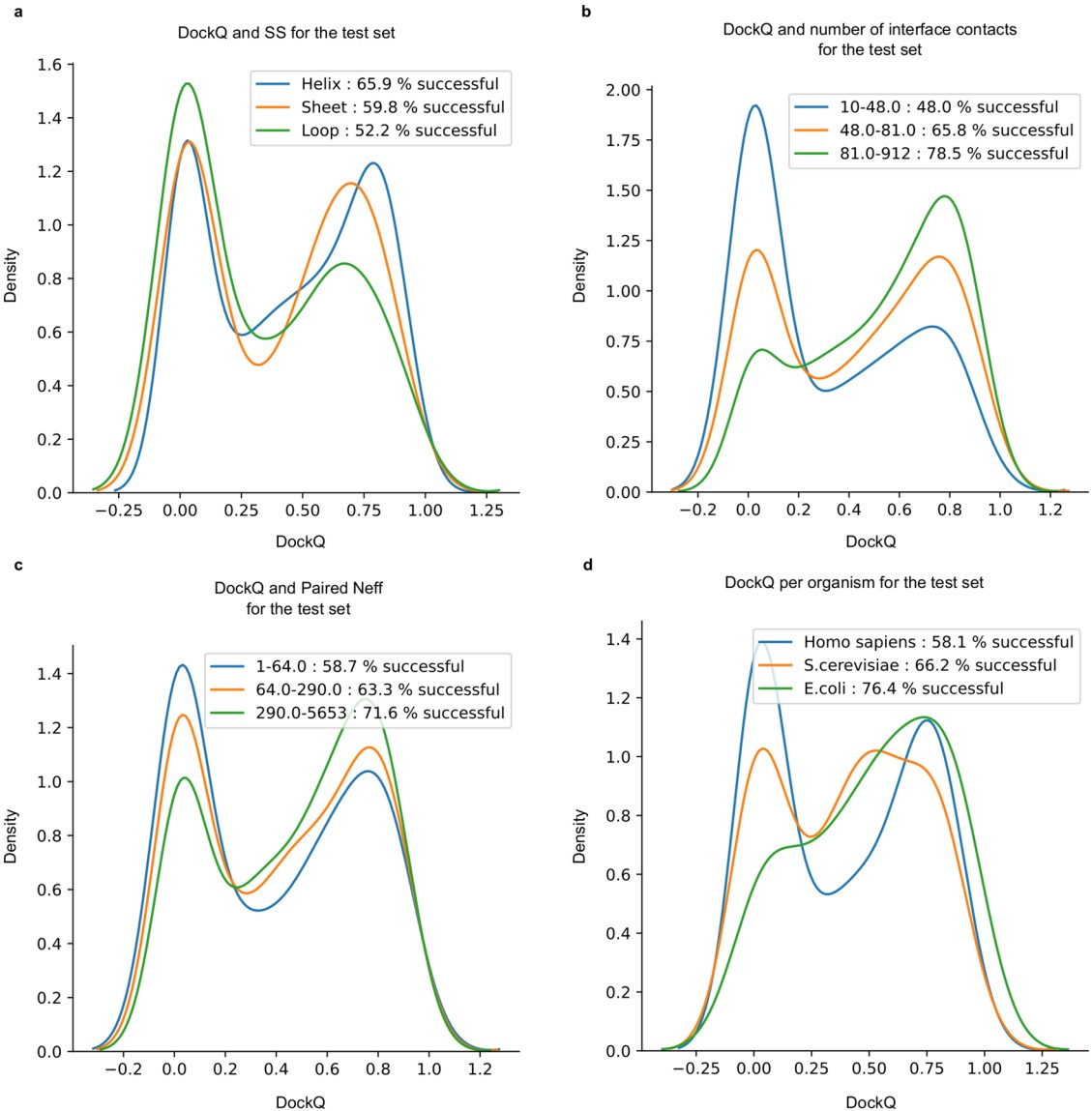

**Fig. 3 DockQ distributions for test dataset (*n* = 1481) tertiles. a** Distribution of DockQ scores for three sets of interfaces with the majority of Helix, Sheet and Coil secondary structures. **b** Distribution of DockQ scores for tertiles derived from the distribution of contact counts in docking model interfaces. **c** Distribution of DockQ scores for tertiles derived from the distribution of Paired MSAs Neff scores. **d** Distribution of DockQ scores for the top three organisms *H. sapiens, S. cerevisiae* and *E. coli*.

performance in *S. cerevisiae* compared to *H. sapiens* suggests a similar relationship between higher and lower order organisms within the same kingdom.

Next, we examine the interfaces. Different secondary structural content of the native interfaces is investigated (Fig. 3a). The highest SR is obtained mainly for helix interfaces (62%), followed by interfaces containing mainly sheets (59%). The loop interface SR of 53% is substantially lower than the others, suggesting that interfaces with more flexible structures are harder to predict. We divide the dataset by interface size, and find that pairs with larger interfaces are easier to predict, as the SR increases from 47 to 74% between the smallest and biggest tertiles (Fig. 3b).

We continue to examine features of the MSAs. First, the impact of the number of non-redundant sequences (Neff) in both paired and AF2 MSAs was analysed. It is clear that the fraction of correctly modelled sequences increases with larger Neff scores (Fig. 3c). Also, paired MSA Neff (Fig. 3c) has a stronger influence on the outcome than the Neff of the AF2 MSAs (Supplementary Fig. 2a). Secondly, the MSA interface signal in the paired MSAs,

measured by the fraction of correct interface contacts using DCA, was analysed. MSAs with stronger interface signals show higher SRs, even if the paired MSAs are used in combination with the AF2 MSAs (Supplementary Fig. 3). This suggests that MSA co-evolutionary signal and, thereby, correct identification of ortholo-gous protein sequences, has a strong impact on the outcome.

**CASP14 and novel proteins without templates**. Chains derived from CASP14 heteromeric targets and chains from PDB com-plexes with no templates are folded in pairs using the presented AF2 pipeline (default AF2 + paired MSAs, ten recycles, m1-10-1 and five differently seeded runs).

For the CASP14 chains, four out of six pairs display a DockQ score larger than 0.23 (SR of 67%). No ranking is necessary in this case, given that all produced docking models for the same chain pair are very similar (the average standard deviation is 0.01 between each set of DockQ scores). An interesting unsuccessful docking is obtained modelling chains from the complex with PDB

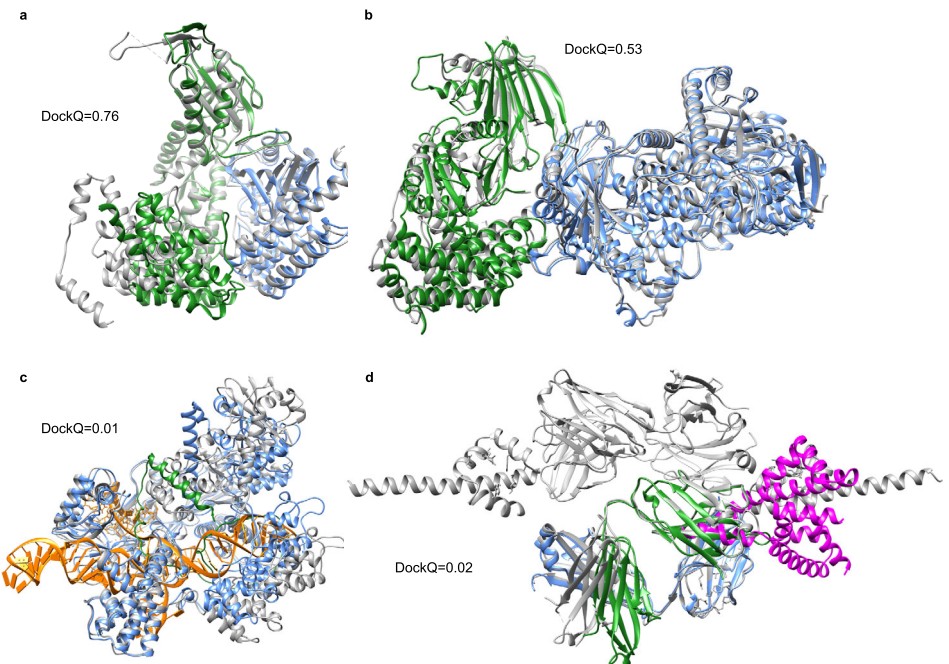

**Fig. 4 Predicted and native structures from the set of novel proteins without templates.** The native structures are represented as grey ribbons. **a** Docking of 7EIV chains A (blue) and C (green) (DockQ = 0.76). **b** Docking of 7MEZ chains A (blue) and B (green) (DockQ = 0.53). **c** Prediction of structure 7EL1 chains A (blue) and E (green) (DockQ = 0.01). The DNA going through chain A is coloured in orange. **d** Docking of 7LF7 chains A (blue) and M (magenta) (DockQ = 0.02) and chains B (green) and M (magenta) (DockQ = 0.02).

ID 6TMM (Supplementary Fig. 4), which are known to form a heterotetramer. In this structure, each chain A is in contact with its partner chain B at two different sites. Both docking configurations (6TMM_A-B and 6TMM_A-D) put the chain in between the two binding sites. The other unsuccessful docking (6VN1_A-H) has an interface of just 19 residue pairs.

The SR for docking the proteins without templates is 50%. Between the five different initialisations, the average difference in the DockQ score is 0.03, and there is no deviation in SR, i.e., ranking did not improve the SR. Two acceptable models are displayed in Fig. 5a (7EIV_A-C]) and B (7MEZ_A-B). More interesting, in one of the incorrect models (7NJ0_A-C], Supplementary Fig. 5), the predictions get the location of both chains correct, but their orientations wrong, resulting in DockQ scores close to 0. For 7EL1_A-E (Fig. 4c), the shorter chain E is not folded correctly, and instead of folding to a defined shape, it is stretched out and inserted within chain A. It occupies the shape of the DNA in the native structure. In the two remaining incorrect models (7LF7_A-M and 7LF7_B-M), Fig. 4d, the chains only interact with a short loop of the M chain, making the docking very difficult and possibly biologically meaningless.

**Identifying interacting proteins**. Using the best separator from the model ranking, the pDockQ, it is possible to distinguish the 3989 non-interacting proteins from *Escherichia coli* and the 1481 truly interacting proteins from the test set with an AUC of 0.87. Another recently published method obtains AUC 0.76 on this set[27]. However, these results are probably overstated since the negative set only contains bacterial proteins, while the positive set is mainly eukaryotic.

To obtain a more realistic estimate, we also include a set of 1705 non-interacting proteins from mammalian organisms[31] combined with the non-interacting proteins from *E. coli*. On this combined set of 1481 interacting and 5694 non-interacting proteins, we obtain an AUC of 0.82 for the average interface plDDT and slightly higher (0.84 and 0.85) for the number of interface contacts and residues, respectively (Fig. 5a). pDockQ

results in an ROC curve with an AUC of 0.87. Importantly, pDockQ provides a better separation at low FPRs, enabling a TPR of 51% at FPR of 1% compared to 27%, 18 and 13% for the interface plDDT, number of interface contacts and residues, respectively. At FPR 5%, the number of interface contacts and residues report TPRs of 49 and 42%, respectively, compared to 43% for the average interface plDDT and 66% for pDockQ. The distribution of the top separators can be seen in Fig. 5b–d.

**Limitations**. Here, we only consider the structures of protein complexes in their heterodimeric state, although each protein chain in these complexes may have homodimer configurations or other higher-order states. It is also possible that the complex itself exists as part of larger biological units, in potentially more complex conformations. Investigating alternative oligomeric states and larger biological assemblies is outside of the scope of this analysis and left for future work.

The study of AF2s ability to separate interacting and non-interacting proteins here contains more extensive data than recent studies[27]. However, to test this separation thoroughly, the data studied here needs to be extended to compare interactions within individual organisms. We leave this extensive analysis to further studies.

There is a big difference between the performance of AF2 on the development and test sets, reporting 39.4% SR vs 57.8% for the AF2 + Paired MSAs. This discrepancy suggests that the performance is highly dependent on the specific interacting partners being predicted. It is not clear what causes this difference as the composition in terms of kingdom, found to be very important (Supplementary Fig. 1b) is similar (54% vs 60% Eukaryotic proteins), the MSAs have similar Neff scores (2699 vs. 2764 on average), the proteins are of similar sizes (222 vs. 203 AAs on average), and the number of residues in the interface is similar (139 vs 120 on average). This leads us to believe that there may be some unknown selection bias in how the sets were chosen. It can be noted that the development is much smaller than the

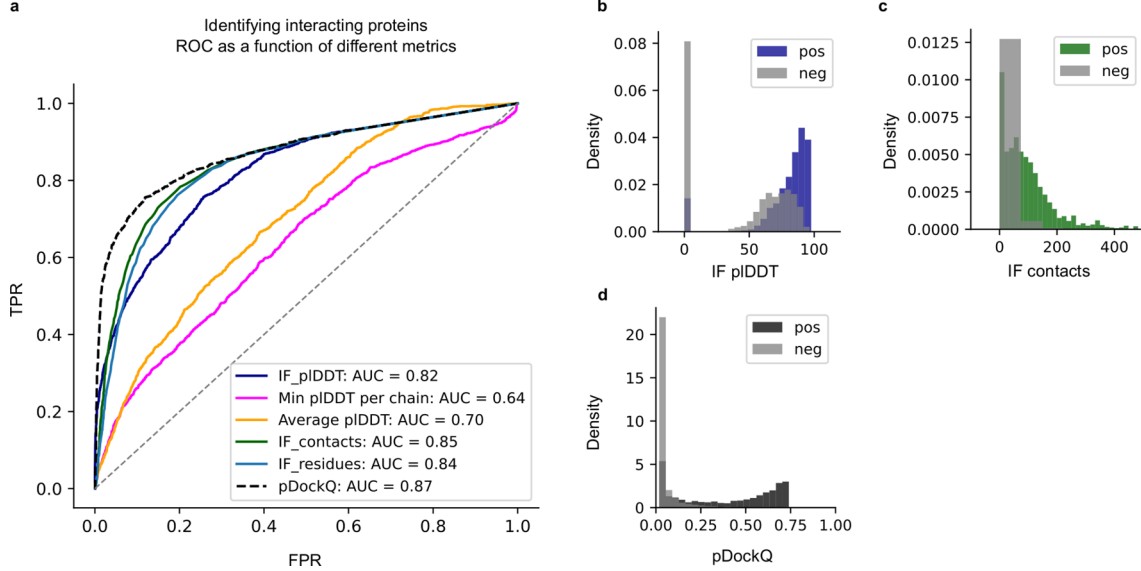

**Fig. 5 Discrimination of interacting ($n = 1481$) and non-interacting ($n = 5694$) proteins. a** The ROC curve as a function of different metrics for discriminating between interacting and non-interacting proteins. IF_plDDT is the average plDDT in the interface, min plDDT per chain is the minimum average plDDT of both chains, average plDDT is the average of the entire complex and IF_contacts and IF_residues are the number of interface residues and contacts respectively. pDockQ is a sigmoidal fit to this with DockQ as the target score, as described above. **b–d** Distribution of the top discriminating features average interface plDDT (**b**), the number of interface contacts (**c**), and **d** the combination of these (IF_plDDT·log(IF_contacts)) and the pDockQ for interacting (non-grey) and non-interacting proteins (grey).

test set though (216 vs 1481 proteins), which is why performance should be assessed on as large non-redundant datasets as possible.

**Findings and future prospects.** Here we show that AlphaFold2[16] (AF2) can predict the structure of many heterodimeric protein complexes, although it is trained to predict the structure of individual protein chains. Even using the default settings, it is clear that AF2 is superior to all other tested docking methods, including other Fold and Dock methods[17,24], methods based on shape complementarity[30,32] and template-based docking. Using optimised MSAs with AF2, we can accurately predict the structure of heterodimeric complexes for an unprecedented SR of 62.7% on a large test set. The SR is higher in *E.coli* (76.4%) than in *H. sapiens* or *S. cerevisiae* (58.1% and 66.2% respectively).

Further, by analysing the predicted interfaces, we can predict the DockQ score[33] (pDockQ) with an average error of 0.1, resulting in the separation of acceptable and incorrect models with an AUC of 0.95. This means that 31% of the models can be called acceptable at a specificity of 99% (or 54% at 90% specificity). Interestingly, no additional constraints are implemented in AF2 to pull two chains in contact, meaning that chain interactions (and subsequently interface sizes) are exclusively determined by the amount of inter-chain signals extracted by the predictor. Assuming that all residues in an interface contribute to the interaction energy could explain why larger interfaces are more likely to be correctly predicted.

We find that the MSA generation process can be sped up substantially at no performance loss (performance increase of 1% SR) by simply fusing MSAs from two HHblits[34] runs on Uniclust30[35] instead of using the MSAs from AF2. Fast MSA generation circumvents the main computational bottleneck in the pipeline. Using pDockQ makes it possible to separate truly interacting from non-interacting proteins with an AUC of 0.87, making it possible to identify 51% of interacting proteins at an error rate of 1%. The pDockQ score discriminates between both model quality and binary interactions. Therefore, the same pipeline can identify if two proteins interact and the accuracy of their structure.

Never before has the potential for expanding the known structural understanding of protein interactions been this large, at such a small cost. There are currently 64,006 pairwise human protein interactions in the human reference interactome[36]. If 31% of these can be predicted at an error rate of 1%, this results in the structure of 19,842 human heterodimeric protein structures. The computational cost to run all of this is ~5 days on an Nvidia A100 system and has since the development of the pipeline presented here, deemed FoldDock, been applied[37].

## Methods

**Development set.** A set of heterodimeric complexes from Dockground benchmark 4[38] is used to develop the pipeline, focusing on the AF2 configuration presented here. This set contains protein pairs, with each chain having at least 50 residues, sharing <30% sequence identity and no crystal packing artefacts. There are 219 protein interactions for which both unbound (single-chain) and bound (interacting chains) structures are available. Unbound chains share at least 97% sequence identity with the bound counterpart and, to facilitate comparisons, non-matching residues are deleted and renumbered to become identical to the unbound counterpart. AF2 MSAs could not be generated for three of the complexes due to memory limitations (1gg2, 2nqd and 2xwb) using a computational node with 128 Gb RAM for the MSA generation and were thus disregarded, resulting in a total of 216 complexes. The dataset consists of 54% Eukaryotic proteins, 38% Bacterial and 8% from mixed kingdoms, e.g., one bacterial protein interacting with one eukaryotic.

**Test set.** We used 1661 protein complexes with known interfaces from a recent study[27] to test the developed pipeline. Here, three large biological assemblies were excluded. These complexes share <30% sequence identity, have a resolution between 1–5 Å and constitute unique pairs of PFAM domains (no single protein pair have PFAM domains matching that of any other pair). Some structures failed to be modelled for various reasons (see limitations of data generation), resulting in a total of 1481 structures. These proteins are mainly from *H. sapiens* (25%), *S. cerevisiae* (10%), *E. coli* (5%) and other Eukarya (30%).

107 of the complexes in the test set lack beta carbons (Cβs), and 50 have overlapping PDB codes with the development set and were therefore excluded. In the MSA generation from AF2, 20 MSAs report MergeMasterSlave errors regarding discrepancies in the number of match states, resulting in a total of 1484 AF2 MSAs. When folding, three of these (5AWF_D-5AWF_B, 2ZXE_B-2ZXE_A and 2ZXE_A-2ZXE_G) report "ValueError: Cannot create a tensor proto whose content is larger than 2GB", leading to a final set of 1481 complexes. DSSP could only be run successfully for 1391 out of the 1481 protein complexes, and we ignored the rest in the analysis.

For RF, 26 complexes produced out of memory exceptions during prediction using a GPU with 40 Gb RAM and were excluded from the RF analyses, leaving 1455 complexes.

For the mammalian proteins from Negatome, seven out of 1733 single chains were redundant according to Uniprot (C4ZQ83, I0LJR4, I0LL25, K4CRX6, P62988, Q8NI70, Q8T3B2), 34 had no matching species in the MSA pairing, 106 produced out of memory exceptions during prediction using a GPU with 40 Gb RAM, 35 gave a tensor reshape error, and 65 complexes were homodimers, leaving 1715 complexes for this set.

**CASP14 set and novel protein complexes.** As an additional test set, we used a set of six heterodimers from the CASP14 experiment. In addition, we extracted eight novel protein complexes deposited in PDB after 15 June 2021, which produced no results for at least one chain in each complex when submitted to the HHPRED web server (version 01-09-2021)[39,40], see Supplementary Table 3. We selected this small set to test the performance on data AF2 is guaranteed not to have seen.

**Non-interacting proteins.** Two datasets of known non-interacting proteins were used, one from the same study as the positive test set[27]. Here, all proteins are from *E. coli*. Two methods were used to identify non-interacting proteins, first a set of proteins with no reported interaction signal in Yeast Two-Hybrid Experiments[41] and secondly complexes whose individual proteins were found in different APMS benchmark complexes[42]. This dataset contains in total 3989 non-interacting pairs.

The second set contains 1964 unique mammalian protein complexes filtered against the IntAct[43] dataset from Negatome[31]. This data deemed "the manual stringent set" contains proteins annotated from the literature with experimental support describing the lack of protein interaction. Some structures in this dataset are homodimers (65) and are therefore excluded, resulting in 1705 structures. Together there are 5694 non-interacting protein complexes.

**AlphaFold2 default MSA generation methodology.** The input to AlphaFold2 (AF2) consists of several MSAs. We used the AF2 MSA generation[16], which builds three different MSAs generated by searching the Big Fantastic Database[44] (BFD) with HHBlits[34] (from hh-suite v.3.0-beta.3 version 14/07/2017) and both MGnify v.2018_12[45] and Uniref90 v.2020_01[46] with jackhmmer from HMMER3[47]. The AF2 MSAs were generated by supplying a concatenated protein sequence of the entire complex to the AF2 MSA generating pipeline in FASTA format. The resulting MSAs will thus mainly contain gaps for one of the two query proteins in each row, as only single chains can obtain hits in the searched databases (Fig. 6). No trimming or gap removal was performed on these MSAs.

**MSA block diagonalization.** In addition to the default AF2 MSA, we generated an additional MSA by simply concatenating diagonally MSAs generated independently from each of the two chains. These MSAs were constructed by running HHblits[34] version 3.1.0 against uniclust30_2018_08[35] with these options:

```
hhblits -E 0.001 -all -oa3m -n 2
```

The concatenation is done by joining side-by-side the two input chains; then sequences from one MSA are added, aligned to the corresponding input chain. Each sequence in the MSA is then elongated with gaps (on the right side if it is the left sequence MSA or the other way around) to reach the length of the two concatenated input chains. The process is then repeated for the other input chain MSA to complete the block diagonalization.

**Paired MSA generation.** In addition to the block diagonalization MSAs, we used a "paired MSA", constructed using organism information, where sequences are matched based on their organism origins[4,21,24] (Fig. 6). The rationale behind using a paired MSA is to identify inter-chain co-evolutionary information. An unpaired MSA has a limited inter-chain signal since the chains are treated in isolation.

The organism information was, using the OX identifier, extracted from the two HHblits MSAs[48]. Next, all hits with more than 90% gaps were removed. From all remaining hits in the two MSAs, the highest-ranked hit from one organism was paired with the highest-ranked hit of the interacting chain from the same organism. Pairing the correct sequences should result in MSAs containing inter-chain co-evolutionary information[27].

**Number of effective sequences (Neff).** To estimate the information in each MSA, we clustered sequences at 62% identity, as described in a previous study[50]. The number of clusters obtained in this way has been used to indicate a $N_{eff}$ value for each MSA.

Unaligned FASTA sequences were extracted from the three AF2 default MSAs. Obtained sequences were processed with the CD-HIT software[51] version 4.7 (http://weizhong-lab.ucsd.edu/cd-hit/) using the options:

```
-c 0.62 -G 0 -n 3 -aS 0.9
```

We calculated the Neff scores separately for paired and AF2 MSAs.

**AlphaFold2.** We modelled complexes using AlphaFold2[16] (AF2) by modifying the script https://github.com/deepmind/alphafold/blob/main/run_alphafold.py to insert a chain break of 200 residues—as suggested in the development of RoseTTAFold[17] (RF). During modelling, relaxation was turned off. We note that performing model relaxation did not increase performance in the AF2 paper[16] and was, therefore, ignored to save computational cost. No templates were used to build structures, as this would not assess the prediction accuracy of unknown structures or structures without sufficient matching templates. Further, AF2 has been shown to perform well for single chains without templates and has reported higher accuracy than template-based methods even when robust templates are available[16].

We supplied four different types of MSAs to AF2: (1) the MSAs generated by using the default AF2 settings, (2) the top paired MSAs constructed using HHblits, described above, (3) both alignments together and finally, (4) the top paired and single-chain MSAs from HHblits to speed up predictions (only for the test set). AF2 was run with two different network models, AF2 model_1 (used in CASP14) and AF2 model_1_ptm, for each MSA. The second model, model_1_ptm, is a fine-tuned version of model_1 that predicts the TMscore[52] and alignment errors[16]. We ran these two different models by using two different configurations. The configurations utilise a varying amount of recycles and ensemble structures. Recycle refers to the number of times iterative refinement is applied by feeding the intermediate outputs recursively back into the same neural network modules. At each recycling, the MSAs are resampled, allowing for new information to be passed through the network. The number of ensembles refers to how many times information is passed through the neural network before it is averaged[16]. The two configurations used are; the CASP14 configuration (three recycles, eight ensembles) and an increased number of recycles (ten) but only one ensembles.

Since structure prediction with AF2 is a non-deterministic process, we generate five models initiated with different seeds. To save computational cost, this was only performed for the best modelling strategy. We rank the five models for each complex by the number of residues in the interface, giving the best result.

**RoseTTAFold.** For comparison, the RoseTTAFold (RF) end-to-end version[17] was run using the paired MSAs with the top hits. The RoseTTAFold pipeline for complex modelling only generates MSAs for bacterial protein complexes, while the proteins in our test set are mainly Eukaryotic. Therefore, we use the paired alignments here. We compare RF with AF2 using the same inputs (the paired MSAs) for both the development and test datasets to provide a more fair comparison, as AF2 searches many different databases to obtain as much evolutionary information as possible when generating its MSAs. To predict the complexes, we use the "chain break modelling" as suggested in RF (https://github.com/RosettaCommons/RoseTTAFold/tree/main/example/complex_modeling) using the following command:

```
predict_complex.py -i msa.a3m -o complex -Ls chain1_length
chain2_length
```

No optimisation of the RF protocol was made here.

**MDockPP.** The docking method MDockPP[30] was run through the provided webserver (https://zougrouptoolkit.missouri.edu/MDockPP/). This docking algorithm is based on fast Fourier transform (FFT). The docking results are assessed using the "in-house" scoring function ITScorePP.

**GRAMM.** For comparison, a rigid-body docking method, GRAMM[32], was used. Here, two protein models are docked using a FFT procedure to generate 340,000 docking poses for each complex. The bound structures extracted from complexes in the test set were used as inputs. This docking generation stage mainly considers the geometric surface properties of the two interacting structures, allowing minor clashes to leave some space for conformational flexibility adjustment. As the bound form of the proteins is used, this should represent an easy case for GRAMM-based docking, and performance drops significantly when unbound structures or models are used[53]. The atom-atom contact energy AACE18 is used to score and rank all poses, as this has been shown to provide better results than shape-complementarity alone[54].

**Template-based docking.** For comparison, a template-based docking protocol[7] referred to as "TMdock" is also adopted. The adopted template library includes 11756 protein complexes obtained from the Dockground database[38] (release 28-10-2020). Monomers from target complexes are structurally aligned with complexes in the supplied libraries (depleted of the target structure PDB ID) in order to identify the best available template structure. The bound form of the template structures was used. TM scores resulting from the alignment of target proteins to each template are averaged and used to score obtained docking models. Alternatively, we refer to "TMdock Interfaces" when targets are structurally aligned only to the template interfaces, defined as every residue with a Cβ atom closer than 12 Å from any Cβ atom in the other chain.

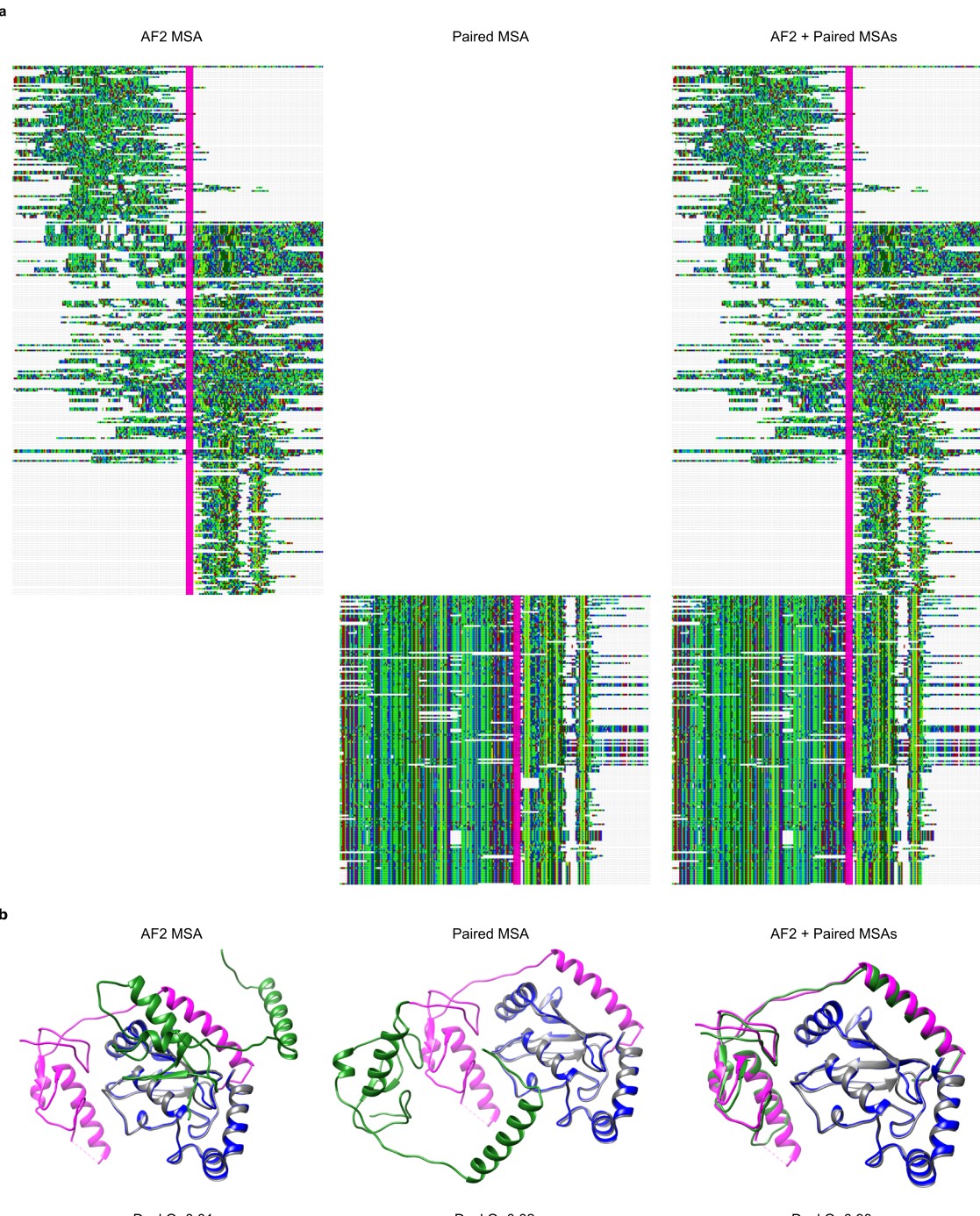

**Fig. 6 Comparison of different MSAs. a** Depiction of MSAs generated by AF2 and the paired version matched using organism information. Both AF and paired representations are sections containing 10% of the sequences aligned in the original MSA. Concatenated chains are separated by a vertical line (magenta). The visualisations were made using Jalview version 2.11.1.4[49]. **b** Docking visualisations for PDB ID 5D1M with the model/native chains A in blue/grey and B in green/magenta using the three different MSAs in (**a**). The DockQ scores are 0.01, 0.02 and 0.90 for AF2, paired, and AF2 + paired MSAs, respectively.

**AlphaFold-multimer**. The simultaneous fold-and-dock program based on the same principles as AF2, AlphaFold-multimer[28], was run with the default settings. These entail creating four different MSAs. Three different MSAs are created by searching Uniref90 v.2020_01[46], Uniprot v.2021_04[48] and MGnify v.2018_12[45] with jackhmmer from HMMER3[47] and one joint is created by searching the Big Fantastic Database[44] (BFD) and uniclust30_2018_08[35] with HHBlits[34] (from hh-suite v.3.0-beta.3 version 14/07/2017).

The results from the Uniprot search are used for MSA pairing and the results from the remaining searches are used to create a block-diagonalized MSA, similar

to the procedures described above. All four MSAs are then used to fold a protein complex. Some complexes failed due to computational limitations, resulting in 1458 out of 1481 complexes successfully folded.

**Scoring models**. The backbone atoms (N, CA and C) were extracted from the predicted AF2 structures (as these are the only predicted atoms in the end-to-end version of RF). The interface scoring program DockQ[33] was then run (without any special settings) to compare the predicted and actual interfaces. This program

compares interfaces using a combination of three different CAPRI[55] quality measures ($F_{nat}$, LRMS, and iRMS) converted to a continuous scale, where an acceptable model comprises a DockQ score of at least 0.23.

**Ranking models**. To analyse the ability of AF2 to distinguish correct models as well as interacting from non-interacting proteins, we analyse the separation between acceptable and incorrect models as a function of different metrics on the development set: the number of unique interacting residues (Cβs from different chains within 8 Å from each other), the total number of interactions between Cβs from different chains (referred to as the number of interface contacts), average predicted lDDT (plDDT) score from AF2 for the interface, the minimum of the average plDDT for both chains and the average plDDT over the whole heterodimer.

We use these metrics as a threshold to build a confusion matrix, where true/false positives (TP and FP respectively) are correct/incorrect docking models which places above the threshold and false/true negatives (FN and TN respectively) are correct/incorrect docking models which scores below the threshold. From the built confusion matrix, we derive the true positive rate (TPR), false positive rate (FPR) defined as:

$$TPR = \frac{TP}{TP + FN} \qquad (1)$$

$$FPR = \frac{FP}{FP + TN} \qquad (2)$$

Then, we calculate TPR and FPR for each possible value assumed by the set of dockings given a single metric and plot TPR as a function of FPR in order to obtain an ROC curve. We compute the area under curve (AUC) for ROC curves obtained for each metric to compare different metrics. The AUC is defined as:

$$AUC = \int_{x=0}^{1} TPR\left(\frac{1}{FPR(x)}\right) dx \qquad (3)$$

The TPR and FPR for different thresholds are used to calculate the fraction of models that can be called correct out of all models and the positive predictive value (PPV). The fraction of acceptable and incorrect models are obtained by multiplying the TPR and FPR with the SR. Multiplying the FPR with the SR results in the false discovery rate (FDR) and the PPV can be calculated by dividing the fraction of acceptable models by the sum of the acceptable and incorrect models. The PPV, FDR and SR are defined as:

$$PPV = \frac{TP}{TP + FP} \qquad (4)$$

$$FDR = 1 - PPV \qquad (5)$$

$$SR = \text{Fraction of predicted models with DockQ} \geq 0.23 \qquad (6)$$

**pDockQ**. As it is not only desirable to know when a model is accurate but also how accurate this model is, we developed a predicted DockQ score, pDockQ. This score is created by fitting a sigmoidal curve (Fig. 2c) using "curve_fit" from SciPy v.1.4.1[56], to the DockQ scores using the average interface plDDT multiplied with the logarithm of the number of interface contacts, with the following sigmoidal equation:

$$pDockQ = \frac{L}{1 + e^{-k(x - x_0)}} + b \qquad (7)$$

where

$$x = \text{average interface plDDT} \cdot \log(\text{number of interface contacts}) \qquad (8)$$

and we obtain $L = 0.724$, $x_0 = 152.611$, $k = 0.052$ and $b = 0.018$.

**Analysis of models**. To analyse the possibility of determining when AF2 can model a complex correctly, we analyse the structures and the MSAs. We investigated: the number of effective sequences (Neff), the secondary structure in the interface annotated using DSSP[57], the length of the shortest chain, the number of residues in the interface and the number of contacts in the interface.

DSSP was run on the entire complexes, and the resulting annotations were grouped into three categories; helix (3-turn helix ($3_{10}$ helix), 4-turn helix (α helix) and 5-turn helix (π helix)), sheet (extended strand in parallel or antiparallel β-sheet conformation and residues in isolated β-bridges) and loop (residues which are not in any known conformation).

In addition, we assess the PPV of the top N interface DCA signals using the paired MSAs. Here, N is the number of true interface contacts (Cβs from different chains within 8 Å from each other). The PPV is therefore the fraction of the top N DCA signals in the interface that are true contacts. The DCA signals are computed using GaussDCA[58].

$$\text{Interface PPV} = \frac{\text{Number of correct contacts among top N interface DCA signals}}{N} \qquad (9)$$

**Computational cost**. To compare the computation required for each MSA, we compared the time it took to generate MSAs for three protein pairs (PDB: 4G4S_P-O, 5XJL_A-2 and 5XJL_2-M), using either the block diagonalization or AF2 protocol. The tests were performed on a computer using 16 CPU cores from an Intel Xeon E5-2690v4.

Fusing the MSAs took 3 s on average per tested complex. It took 7884 s for generating the AF2 MSAs, the single-chain searches took 338 s on average and the pairing 2 s. The pairing and fusing are thereby negligible compared to searching, resulting in a speedup of 24 times for the hhblits searches. In comparison, folding using the m1-10-1 strategy took 191 s on average for these pairs.

**Reporting summary**. Further information on research design is available in the Nature Research Reporting Summary linked to this article.

## Data availability
The raw data used in this study, including multiple sequence alignments and predicted PDB files, are available in the figshare from Science for Life laboratory under accession code 16866202.v1. All other data supporting the findings of this study are available within the article and its supplementary information files. The results used to produce all figures can be found in the supplementary information. Additional information and relevant data will be available from the corresponding author upon reasonable request.

## Code availability
All code to run FoldDock and reproduce the analysis here can be obtained here https://gitlab.com/ElofssonLab/FoldDock (commit 2e4c96aa352338976260ece0646ceaaa75392dec) under the Apache License, Version 2.0.

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

## Acknowledgements

We thank Petras Kundrotas for supplying the new heterodimeric proteins without templates in the PDB. We also thank Liming Qiu and Xiaoqin Zou for their help with running their docking program MDockPP in a timely manner. Financial support: Swedish Research Council for Natural Science, grant No. VR-2016-06301 and Swedish E-science Research Center. Computational resources: Swedish National Infrastructure for Computing, grants: SNIC 2021/5-297, SNIC 2021/6-197 and Berzelius-2021-29. All financial support and computational resources were received by A.E.

## Author contributions

P.B. and G.P. performed the studies; all authors contributed to the analysis. P.B. wrote the first draft of the manuscript; all authors contributed to the final version. A.E. obtained funding.

## Funding

## Competing interests

The authors declare no competing interests.
