## [Peer Review File · Nature Communications]

Improved prediction of protein-protein interactions using AlphaFold2REVIEWER COMMENTS

Reviewer #1 (Remarks to the Author):

The manuscript demonstrates a way to improve the accuracy of AF2 predicted hetero-dimeric protein complexes by optimising the multiple sequence alignment. It also tests if it is possible to distinguish interacting from non-interacting pairs by analysis the proposed interfaces. The code is available as open source licensed under Apache2 and distributed via GitLab.

Authors state that flexibility can limit the accuracy of rigid-body docking and correctly point that flexible docking too slow and inaccurate for large scale applications, however it should also be mentioned that these are not the only possible routes and many docking software have a semi-flexible approach, and can, to some degree, consider the flexibility during docking.

It is also stated that (L94) "We find that the results in terms of successful docking using AF2 are superior to all other docking methods." However the authors have shown a comparison using two, out of dozens of possible docking methods, including others that could factor interface flexibility and thus more suitable for fold and dock comparison. This statement could be rewritten to best reflect the observed results.

In the light of open science and FAIR data, authors should make all the data used in this manuscript available to the community (via Zenodo, SBGrid databases, etc).

Please indicate on L211 and L251 the exact commit which relates to this edit.

For GRAMM, the bound forms of the test set are used as input, which would represent an "easy scenario" for docking. This is an adequate approach since the objective of the comparison is not to evaluate how well this rigid-body method would be able to model the flexibility. For template-based docking, it is unclear if the bound or unbound forms were used.

The scoring for the AF2 structures is done only over the backbone atoms, it is not clear if this same approach is used for the comparison with GRAMM and TMDock. The choice completely excluding the side-chains instead of adding them to the only method that does not predict it, thus reducing the resolution of solutions seems to go against the overall goal of the manuscript which is the increase of accuracy, the rationale for this should be explicit in the text.

It is not specified if the cutoff for the frequency of the native contacts was changed, which could indicate a sub-evaluation of this metric since it might only be capturing backbone-to-backbone contacts. The authors report the DockQ score, however it would be beneficial to have a table containing the fnat, lrms, irms and DockQ of the scored models (could be deposited together with the rest of the data).

Two small details that could increase readability is to add the short names in L350 and a horizontal line with the DockQ cutoff on Figure 2, but not entirely necessary.

I have no comments on the sections "Distinguishing acceptable from incorrect models", "model variation and ranking", "Bacterial protein pairs with large interfaces and many homologs are easier to predict" and "CASP14 and novel proteins without templates"; the results are presented clearly and well discussed.

Based on the results observed for identifying interacting proteins, could the authors propose a "cutoff" that can be used by researchers to judge if a given pair is a true interaction? The text seems to imply that this was the direction of this analysis. The number of interface contacts and number of residues in the interface have a higher AUC, it would be interesting to analyse the identification in subsets of differently sized interfaces (or expand the discussion to include this observation).

L528 should be rewritten to "the tested docking methods" since the authors do not present a through comparison with many different software.

The fast MSA generation presented in this manuscript is a noteworthy result and the differentiation between true-interacting and non-interacting proteins (given its described limitations) sets a solid base for further studies in this direction.

Reviewer #2 (Remarks to the Author):

This paper evaluates the performance of the currently available implementation of AlphaFold 2 (AF2) DL model in predicting the 3D structure of heteromeric protein complexes and investigates quantitative measures for discriminating between AF2 predicted structures corresponding to correct versus incorrect predictions.

This AF2 model was trained on individual protein structures, and shown to 1) outperform competing methods in ab-initio structure predictions of single protein chains as well as for template-based predictions, in the CASP14 challenge, 2) produce protein models rivaling in accuracy with experimentally determined structures, 3) achieve this performance for individual domains in multi-domain proteins, or for individual subunits of larger oligomers, without explicitly taking into account the domain architecture or quaternary structure of the protein. The latter achievement suggested that this AF2 DL model captures information that transcended the fold of individual proteins and may be exploited for predicting the 3D structure of multi-domain proteins and protein complexes. Several follow up studies (most not (yet) peer reviewed) showed indeed that that providing pseudo-multimer inputs to the single-chain AF2 model (joining two protein sequence with a gap insertion or a flexible linker) often yields successful predictions of multimer interactions. The present study is part of these efforts.

Using protein complexes from the Dockground benchmark 4 (the development set) the authors test various settings for the inference procedure in the available AF2 implementation and select the settings that yield the best performance on this set (without templates), as measured by the fraction of recalled complexes of acceptable quality or better (DockQ >0.23).

Evolutionary signals derived from multiple sequence alignments (MSA) (informing on residue-residue interactions) are an important component of the single-chain AF2 model. Expecting this component to also play a key role in the effective generalization of the model to the prediction of complexes, the authors test different methods for generating the MSAs. The combination of MSAs generated by 2 methods, the default AF2 MSA generation method (producing MSAs containing gaps for one of the two query proteins in each row) and the paired MSAs method (the highest-ranked hit for chain A from one organism is paired with the highest-ranked hit of chain B from the same organism), is shown to perform best in predictions for the development set.

Using the optimized inference protocol and MSAs generation, AF2 performance is evaluated for the task of predicting complexes from a test set (1481 complexes with known interfaces from Green et al.) and on CASP14 targets, again as measured by the fraction of recalled complexes of acceptable quality or better. The best AF2 protocol evaluated for the test set achieves about 60% of correctly recalled of the complexes, a roughly similar performance to or sometime lower than those cited in other works (using different test sets).

In a more controversial part of the study AF2 performance is compared to that obtained for the same test set using a single 'ab-initio' rigid-body docking procedure (GRAMM), taking as input the bound conformations of the interacting subunits. A comparison was also made to results obtained using two version of so-called template-based docking (TMdock, and TMdock interface). Both the ab-initio and TMdock procedures are shown to achieve significantly lower recall rates (~21% for GRAMM, and 34-35% for TM-dock). A very low recall fraction (~10%), the lowest overall, is obtained using the RoseTTAFold (RF) end-to-end version.

Based on these comparisons the authors claim that the AF2 outperforms the other approaches, and in particular docking protocols by a large margin. Or such claim cannot be made on the basis of a comparison with the performance of a single docking procedure, which is furthermore not representative of the field as it stands now. Indeed, a number of other docking procedures (also available as servers) such as CLUSPRO, LZERD, MDCKPP, tackle conformational flexibility at some level and systematically outperform GRAMM in more recent blind prediction challenges, including the CASP14 assembly prediction. The computational costs of some of these algorithms may be higher than for pure rigid body docking algorithms like GRAMM, but this can hardly be used to justify the analysis, and the conclusions drawn.

The study also evaluates the ability to segregate correct models from incorrect ones in AF2 structures predicted for a test dataset of complexes that include both positive and negative examples (respectively, protein complexes with experimentally determined structures, and protein pairs assumed not to interact). Analyzing ROCs as a function of various quantitative measures, the pIDDT score computed by AF2 is found to perform on par (AUC: 85%) with measures that directly correlated with interface size, such as the total number of inter-subunits residue-residue interactions, and the total number of interface residues (AUC : 86%). This is not surprising and agrees with earlier finding that stable complexes, which form larger interfaces are easier to predict correctly, than transient complexes, which form smaller interfaces, a property picked up by pIDDT, but not by pDDT, which evaluate the predicted model accuracy for the entire complex and not only for the residues at the binding interface.

Overall, this study provides useful information on how to adapt the single-chain AF2 protocol for the prediction of protein hetero complexes, more particularly on how to improve the signal extracted from MSAs for this purpose. On the other hand, the comparisons with the performance of ab-initio docking and template-based docking are suboptimal and do not justify the general claims made here. The comparison with RF may likewise be suboptimal, since no parameter optimization was performed for the RF procedure. On the whole, the text is very technical and offers only limited insightful discussion.

Lastly, one may question the overall impact of the presented work in view of the recent publication by the DeepMind team describing AlphaFold-Multimer (<https://doi.org/10.1101/2021.10.04.463034>), an AlphaFold deep learning model trained on complexes of known structure, and shown to outperform the single-chain AlphaFold model by 11 to 25 percentage points. Moreover, DeepMind recently announced that the AlphaFold-Multimer code is about to be publicly released.

Specific comments

Introduction:

-The authors define ab-initio docking methods as methods relying solely on shape complementarity. Initially this was indeed the case, but methods have evolved since then to optimize not only shape complementarity but also various additional energetic contributions. State of the art docking algorithms are also capable of modeling limited conformational adjustments.

Methods section

-The number of complexes in the Dockground benchmark 4 (the development dataset) and their species composition should be provided

- A short description in terms of species composition should be provided of the test dataset of positive example (complexes of known interfaces) from Green et al.

-The description of the datasets of non-interacting proteins is confusing. Two paragraphs mention the Negatome DB as the source of negative examples: one is part of the section describing the test dataset and lists a total 1715 pairs of non-interacting proteins. The second appears in the section on non-interacting proteins, and lists 1705 pairs from the Negatome DB. The same section also mentions another dataset of non-interacting proteins (also from Green et al.). What was the species compositions of these datasets? In which of the analyses on the segregation of correct versus incorrect complexes, were these datasets used?

-The notoriously difficult problem of defining non-interacting proteins should be given some consideration, as some of the cited criteria are problematic. F.e. Y2H screens are well known for their high rate of false negatives, e.g. proteins that interact in-vivo, whose interaction is not detected using Y2H for various reasons. Likewise, considering proteins from different well annotated APMS complexes as non-interacting may also be misleading as many proteins are found to be part of multiple APMS complexes. BTW: ref 31 seems to be incorrect

- How is the number of effective sequences computed. Relying on the literature citation (ref 62) is not satisfactory.

Results

- Is the superior performance of the combined AF2 + paired MSA's really only a consequence of the larger size of the resulting MSA ?

- As already mentioned above, the comparison of the AF2 performance with that of a single docking algorithm (GRAMM), which moreover doesn't represent the state of the art, is unfair. It may indeed be the case that a fairer comparison may prove the authors right, but this needs to be based on a valid evidence.

-The template-based docking procedure used in this study seems to be quite different from the procedure the authors refer to (ref 7). Here, it seems, target complexes from the test dataset are structurally aligned either to the backbone of the full template complex, or only to the template interface residues. Or in blind predictions, where the structure of the target complex is unknown, template-based 'docking' involves aligning the sequences of each protein of the target complex to that of its homolog in the template complex and going on from there.

-The legend of Figure 3 reads: 'ROC curve as a function of different metrics for the development dataset (first run). But the text refers to Figure 3 as representing the results for the test set. Indeed, the ROC of Figure 3A represents a plot of the TPR versus FPR and requires scoring & ranking predicted structures for both the TP and TN examples, which the text does not describe for the development set. What then is the difference between the ROCs in Figures 3A and 6A ?

ANSWER TO REVIEWERS COMMENTS

We want to thank the reviewers for the number of fascinating discussion points they provided us, which allowed us to considerably improve our manuscript. We here provide a detailed answer to all remarks. The main new additions are

1. The comparison in performance with the docking method MDockPP and AlphaFold-multimer
2. Addition of a continuous metric for model quality assessment, the predicted DockQ score, pDockQ.
3. We have changed the naming of the “fused” MSAs to “block diagonalisation” to better reflect the phrasing being used by others.

REVIEWER COMMENTS

Reviewer #1 (Remarks to the Author):

The manuscript demonstrates a way to improve the accuracy of AF2 predicted hetero-dimeric protein complexes by optimising the multiple sequence alignment. It also tests if it is possible to distinguish interacting from non-interacting pairs by analysis the proposed

interfaces. The code is available as open source licensed under Apache2 and distributed via GitLab.

Authors state that flexibility can limit the accuracy of rigid-body docking and correctly point that flexible docking too slow and inaccurate for large scale applications, however it should also be mentioned that these are not the only possible routes and many docking software have a semi-flexible approach, and can, to some degree, consider the flexibility during docking.

We have added a point about this in the introduction:

“There also exists semi-flexible docking approaches that are more computationally feasible and can consider flexibility to some degree during docking.”

and also included an analysis using the top-performing available server method from CASP14-CAPRI MDOCKPP -

It is also stated that (L94) "We find that the results in terms of successful docking using AF2 are superior to all other docking methods." However the authors have shown a comparison using two, out of dozens of possible docking methods, including others that could factor interface flexibility and thus more suitable for fold and dock comparison. This statement could be rewritten to best reflect the observed results.

To address this issue, we have added a comparison using the top-performing available server method from CASP14-CAPRI, MDOCKPP. and as expected AF2 clearly outperforms that server (which is on par with GRAMM that we included in our original analysis)

In the light of open science and FAIR data, authors should make all the data used in this manuscript available to the community (via Zenodo, SBGrid databases, etc).

We have made all data available through Figshare:

<https://doi.org/10.17044/scilifelab.16866202.v1> and the results used to produce all figures can be found in the supplementary information.

Please indicate on L211 and L251 the exact commit which relates to this edit.

We have added this information in the code availability section

(<https://gitlab.com/ElofssonLab/FoldDock>, commit 2e4c96aa352338976260ece0646ceaaa75392dec).

For GRAMM, the bound forms of the test set are used as input, which would represent an "easy scenario" for docking. This is an adequate approach since the objective of the comparison is not to evaluate how well this rigid-body method would be able to model the flexibility. For template-based docking, it is unclear if the bound or unbound forms were used.

We have tried to clarify that the bound forms have been used under the “template-based docking” section - as these were the only available forms for the test set.

The scoring for the AF2 structures is done only over the backbone atoms, it is not clear if this same approach is used for the comparison with GRAMM and TMdock. The choice completely excluding the side-chains instead of adding them to the only method that does not predict it, thus reducing the resolution of solutions seems to go against the overall goal of the manuscript which is the increase of accuracy, the rationale for this should be explicit in the text.

The reason for using only backbone atoms is to make the comparison fair towards RoseTTAFold, since this method only predicts the N, CA and C atoms. Adding these atoms to RoseTTAFold creates another conundrum as this will be highly dependent on the program used to add these atoms. This would further allow an argument regarding that the performance of RoseTTAFold is related to the performance of the program used for the atom addition.

Regardless, we have rerun the analysis using all predicted atoms from AlphaFold2, while leaving the RoseTTAFold analysis as it is to not introduce further bias and due to the fact that RoseTTAFold actually only predicts backbone atoms. For GRAMM and TMdock we already used all atoms, which is why these SRs remain the same. We have also added the best server docking methods from CASP14-CAPRI - MDOCKPP - evaluated using all atoms as well.

Note that this has also changed the results on the development set, although the relationships between all modeling strategies remain the same (Table 1). For the pdb files with a discrepancy in the amount of atoms towards the AlphaFold2 predictions, due to e.g. incomplete residues (1.6% of structures) we have used the method from before, only analysing the backbone atoms (N,CA,C).

The results in the test sets are improved from 59% to **63%** SR using the best method and ranking (Figure 1). Now, the interface contacts outperform the interface pIDDT in model ranking but the difference is small.

During revision we realised that it is possible to combine these metrics in a simple multiplication, resulting in an AUC of 0.95 and a significant improvement at low FPRs compared to using each ranking metric alone (Figure 2A). We also create a continuous score from this combined metric to provide users with a straightforward measure of interface quality, the predicted DockQ (pDockQ). This score estimates the DockQ score by applying a simple sigmoidal curve fit, resulting in an average error of 0.1 in DockQ score (see pDockQ in methods). We do believe this is a useful measure.

It is not specified if the cutoff for the frequency of the native contacts was changed, which could indicate a sub-evaluation of this metric since it might only be capturing backbone-to-backbone contacts. The authors report the DockQ score, however it would be beneficial to have a table containing the fnat, lrms, irms and DockQ of the scored models (could be deposited together with the rest of the data).

No cutoff was made, the DockQ program was run in its default mode. We have added a note about this under “scoring models”. The DockQ score is a combination of fnat, lrms and irms, creating a continuous score that considers multiple interface metrics. DockQ is also the continuous score reported for evaluations such as CASP14-CAPRI (<https://onlinelibrary.wiley.com/doi/epdf/10.1002/prot.26222>) why we do think it is sufficient to include it and adding more measures would just confuse the reader. Obviously for small differences between methods alternative methods might provide small different rankings, but here we discuss more than a doubling of the number of acceptable models, and so large differences would not change using any measure.

Two small details that could increase readability is to add the short names in L350 and a horizontal line with the DockQ cutoff on Figure 2, but not entirely necessary.

We thank you for the suggestion. We have reduced the information in Figure 2 (now Figure 1) since more comparisons have been added (MDockPP), making the figure crowded. We now report the main results in this figure in the form of boxplots and more detailed results for all comparisons in Table 2.

I have no comments on the sections "Distinguishing acceptable from incorrect models", "model variation and ranking", "Bacterial protein pairs with large interfaces and many homologs are easier to predict" and "CASP14 and novel proteins without templates"; the results are presented clearly and well discussed.

We thank you for this kind statement.

Based on the results observed for identifying interacting proteins, could the authors propose a "cutoff" that can be used by researchers to judge if a given pair is a true interaction? The text seems to imply that this was the direction of this analysis.

With the introduction of pDockQ this has become much easier. A reasonable cutoff can be deduced depending on modelling necessities according to statistics indicated in Table S4. Such a cutoff is e.g. pDockQ=0.5 for a PPV of 0.9 and using the default DockQ cutoff for an acceptable model (0.23) results in a PPV of 0.75. However, these estimates are probably conservative as a few of the false positives are likely to be alternative binding sites.

The number of interface contacts and number of residues in the interface have a higher AUC, it would be interesting to analyse the identification **in subsets of differently sized interfaces** (or expand the discussion to include this observation).

We analyse this in Figure 3B, where we show that true interfaces that are larger have higher DockQ scores, meaning that more of these models will be called successful. This is explained in the text as “We divided the dataset by the size of the interface, and it is clear that pairs with larger interfaces are easier to predict, as the SR increases from 48 to 79% between the smallest and biggest tertiles (Figure 3B).”

L528 should be rewritten to "the tested docking methods" since the authors do not present a through comparison with many different software.

We have changed this to “What is most striking is that AF2 outperforms all other tested docking methods by a large margin.”

The fast MSA generation presented in this manuscript is a noteworthy result and the differentiation between true-interacting and non-interacting proteins (given its described limitations) sets a solid base for further studies in this direction.

We agree and thank you for noticing this result that has significant importance for large-scale practical applications.

Reviewer #2 (Remarks to the Author):

This paper evaluates the performance of the currently available implementation of AlphaFold 2 (AF2) DL model in predicting the 3D structure of heteromeric protein complexes and investigates quantitative measures for discriminating between AF2 predicted structures corresponding to correct versus incorrect predictions.

This AF2 model was trained on individual protein structures, and shown to 1) outperform competing methods in ab-initio structure predictions of single protein chains as well as for template-based predictions, in the CASP14 challenge, 2) produce protein models rivaling in accuracy with experimentally determined structures, 3) achieve this performance for individual domains in multi-domain proteins, or for individual subunits of larger oligomers, without explicitly taking into account the domain architecture or quaternary structure of the protein. The latter achievement suggested that this AF2 DL model captures information that transcended the fold of individual proteins and may be exploited for predicting the 3D structure of multi-domain proteins and protein complexes. Several follow up studies (most not (yet) peer reviewed) showed indeed that that providing pseudo-multimer inputs to the single-chain AF2 model (joining two protein sequence with a gap insertion or a flexible linker) often yields successful predictions of multimer interactions. The present study is part of these efforts.

Using protein complexes from the Dockground benchmark 4 (the development set) the authors test various settings for the inference procedure in the available AF2 implementation and select the settings that yield the best performance on this set (without templates), as measured by the fraction of recalled complexes of acceptable quality or better (DockQ >0.23).

Evolutionary signals derived from multiple sequence alignments (MSA) (informing on residue-residue interactions) are an important component of the single-chain AF2 model. Expecting this component to also play a key role in the effective generalization of the model to the prediction of complexes, the authors test different methods for generating the MSAs. The combination of MSAs generated by 2 methods, the default AF2 MSA generation method (producing MSAs containing gaps for one of the two query proteins in each row) and the paired MSAs method (the highest-ranked hit for chain A from one organism is paired with the highest-ranked hit of chain B from the same organism), is shown to perform best in predictions for the development set.

Using the optimized inference protocol and MSAs generation, AF2 performance is evaluated for the task of predicting complexes from a test set (1481 complexes with known interfaces from Green et al.) and on CASP14 targets, again as measured by the fraction of recalled

complexes of acceptable quality or better. The best AF2 protocol evaluated for the test set achieves about 60% of correctly recalled of the complexes, a roughly similar performance to or sometime lower than those cited in other works (using different test sets).

Although some papers have reported these high numbers, we are not aware of any methods that can perform this well on the first ranked models for a set of unbound protein models. We have added the current state-of-the-art server method from CASP14-CAPRI, MDockPP, which performs significantly worse than any AlphaFold2 configuration (Figure 1).

In a more controversial part of the study AF2 performance is compared to that obtained for the same test set using a single 'ab-initio' rigid-body docking procedure (GRAMM), taking as input the bound conformations of the interacting subunits. A comparison was also made to results obtained using two version of so-called template-based docking (TMdock, and TMdock interface). Both the ab-initio and TMdock procedures are shown to achieve significantly lower recall rates (~21% for GRAMM, and 34-35% for TM-dock). A very low recall fraction (~10%), the lowest overall, is obtained using the RoseTTAFold (RF) end-to-end version.

Based on these comparisons the authors claim that the AF2 outperforms the other approaches, and in particular docking protocols by a large margin. Or such claim cannot be made on the basis **of a comparison with the performance of a single docking procedure, which is furthermore not representative of the field as it stands now. Indeed, a number of other docking procedures (also available as servers) such as CLUSPRO, LZERD, MDOCKPP, tackle conformational flexibility at some level and systematically outperform GRAMM in more recent blind prediction challenges, including the CASP14 assembly prediction.** The computational costs of some of these algorithms may be higher than for pure rigid body docking algorithms like GRAMM, but this can hardly be used to justify the analysis, and the conclusions drawn.

We agree that GRAMM might not be state-of-art. To obtain a more complete comparison, we have therefore included MDOCKPP (the top performing server method from CASP14-CAPRI). As can be seen in Table2 the performance of MDOCKPP and GRAMM are very similar.

The study also evaluates the ability to segregate correct models from incorrect ones in AF2 structures predicted for a test dataset of complexes that include both positive and negative examples (respectively, protein complexes with experimentally determined structures, and protein pairs assumed not to interact). Analyzing ROCs as a function of various quantitative measures, the pIDDT score computed by AF2 is found to perform on par (AUC: 85%) with measures that directly correlated with interface size, such as the total number of inter-subunits residue-residue interactions, and the total number of interface residues (AUC : 86%). This is not surprising and agrees with earlier finding that stable complexes, which form larger interfaces are easier to predict correctly, than transient complexes, which form smaller

interfaces, a property picked up by pIDDT, but not by pDDT, which evaluate the predicted model accuracy for the entire complex and not only for the residues at the binding interface.

We have rerun all analysis using all atoms for scoring, as suggested by another reviewer. The results are improved from 59% to **63%** SR using the best method and ranking (Figure 1). Now, the interface contacts outperform the interface pIDDT in model ranking. More importantly now we combine these metrics into the predicted DockQ (pDockQ) score, by using a simple multiplication, resulting in an AUC of 0.95 and a significant improvement at low FPRs compared to using each ranking metric alone (Figure 2A). The creation of the continuous pDockQ score from this combined metric provides users with a straightforward measure of interface quality. This score estimates the DockQ score by applying a simple sigmoidal curve fit, resulting in an average error of 0.1 in DockQ score (see pDockQ in methods).

Overall, this study provides useful information on how to adapt the single-chain AF2 protocol for the prediction of protein hetero complexes, more particularly on how to improve the signal extracted from MSAs for this purpose. On the other hand, the comparisons with the performance of ab-initio docking and template-based docking are suboptimal and do not justify the general claims made here. The comparison with RF may likewise be suboptimal, since no parameter optimization was performed for the RF procedure. On the whole, the text is very technical and offers only limited insightful discussion.

We have made it clear that the RF protocol was not optimised in any way in the RoseTTAFold methods section: “No optimisation of the RF protocol was made here. “ It is possible that using another scheme for alignments would result in better results. However, we are not able to achieve this, and it can also be highlighted that the Baker group (see <https://www.biorxiv.org/content/biorxiv/early/2021/09/30/2021.09.30.462231.full.pdf>) choose to use AlphaFold2 to model eukaryotic complexes, indicating that even the developers of RoseTTAFold achieve better results using AF2.

As mentioned above, we have also added MDOCKPP to obtain a more complete docking comparison.

Lastly, one may question the overall impact of the presented work in view of the recent publication by the DeepMind team describing AlphaFold-Multimer (<https://doi.org/10.1101/2021.10.04.463034>), an AlphaFold deep learning model trained on complexes of known structure, and shown to outperform the single-chain AlphaFold model by 11 to 25 percentage points. Moreover, DeepMind recently announced that the AlphaFold-Multimer code is about to be publicly released.

We have indeed noticed this new work by DeepMind. However, this was not available at the time of submission and it is stated that this work is still in progress at their github (“*This represents a work in progress and AlphaFold-Multimer isn't expected to be as stable as our monomer AlphaFold system.*”). We have added a comparison with AlphaFold-multimer anyhow, resulting in SR=72%, 9% better than our updated FoldDock pipeline (63%). We note that AlphaFold-multimer is developed using the same data as the test set here, which makes a direct comparison difficult.

Specific comments

Introduction:

-The authors define ab-initio docking methods as methods relying solely on shape complementarity. Initially this was indeed the case, but methods have evolved since then to optimize not only shape complementarity but also various additional energetic contributions. We have reformulated the introduction to properly reflect this issue stating in a more general manner that it is possible to “select the correct docking through a scoring function” and that there are other more flexible methods:

“A possible compromise is represented by semi-flexible docking approaches¹³ that are more computationally feasible and can consider flexibility to some degree during docking.”

State of the art docking algorithms are also capable of modeling limited conformational adjustments.

We have added a note about this in the introduction:

“A possible compromise is represented by semi-flexible docking approaches¹³ that are more computationally feasible and can consider flexibility to some degree during docking.”

Methods section

-The number of complexes in the Dockground benchmark 4 (the development dataset) and their species composition should be provided

These are already provided in the methods section. There are in total 216 protein complexes in the development set and “The dataset consists of 54% Eukaryotic proteins, 38% Bacterial and 8% from mixed kingdoms, e.g. one bacterial protein interacting with one eukaryotic.” (see methods under development set). The exact kingdom/species for each protein is available in the supplementary data.

- A short description in terms of species composition should be provided of the test dataset of positive example (complexes of known interfaces) from Green et al.

This information is provided in the Methods sections under “Test set”: *“These proteins are mainly from H. Sapiens (25%), S. Cerevisiae (10%), E.coli (5%) and other Eukarya (30%).”* The exact species/kingdom is provided in the supplementary data.

-The description of the datasets of non-interacting proteins is confusing. Two paragraphs mention the Negatome DB as the source of negative examples: one is part of the section describing the test dataset and lists a total 1715 pairs of non-interacting proteins. The second appears in the section on non-interacting proteins, and lists 1705 pairs from the Negatome DB. The same section also mentions another dataset of non-interacting proteins (also from Green et al.). What was the species compositions of these datasets? In which of the analyses on the segregation of correct versus incorrect complexes, were these datasets used?

We are sorry that this was confusing. Two datasets have been used as negative controls for the purpose of separating interacting and non-interacting proteins. The species composition and origin of these datasets is described in methods under “non-interacting proteins”:

Two datasets of known non-interacting proteins were used, one from the same study as the positive test set²⁷. Here, all proteins are from E.coli. The second set contains 1964 unique mammalian protein complexes filtered against the IntAct³⁵ dataset from Negatome³⁶.

-The notoriously difficult problem of defining non-interacting proteins should be given some consideration, as some of the cited criteria are problematic. F.e. Y2H screens are well known for their high rate of false negatives, e.g. proteins that interact in-vivo, whose interaction is not detected using Y2H for various reasons. Likewise, considering proteins from different well annotated APMS complexes as non-interacting may also be misleading as many proteins are found to be part of multiple APMS complexes. BTW: ref 31 seems to be incorrect

It is true the experiments supporting the annotation of interacting and non-interacting proteins may contain errors (and we have indeed seen that for some example). Still, this data is currently the best available and is used in various studies, including the recently published one we compare with (<https://www.nature.com/articles/s41467-021-21636-z>). To address some of the problems in these comparisons we combine two datasets, one of bacterial and one of mammalian origin, showing that the performance remains (AUC=0.87), giving support to the possibility of separating truli interacting from non-interacting proteins across kingdoms.

Regarding REF31, we thank you for such a careful consideration and refer to that this is the reference requested to be cited by the authors of the HHpred webserver (see <https://toolkit.tuebingen.mpg.de/tools/hhpred>).

- How is the number of effective sequences computed. Relying on the literature citation (ref 62) is not satisfactory.

We have rephrased the Neff methods section to clarify this point:

“To estimate the information in each MSA, we clustered sequences at 62% identity, as described in a previous study⁴⁵. The number of clusters obtained in this way has been used to indicate a N_{eff} value for each MSA.
“

Results

- Is the superior performance of the combined AF2 + paired MSA's really only a consequence of the larger size of the resulting MSA ?

Understanding exactly what information is used by AF2 is difficult. However, what is clear is that it is possible in some cases to use non-paired (often referred to as block diagonalisation) alignments and still obtain good predictions, while in other cases the “paired” alignments work well (and for some none or both work). It is also clear that in general, pairs with a stronger co-evolutionary signal perform better on average.

Both methods rely on HHblits and Uniclust30 to generate MSAs, so the two different MSAs (AF default and paired) are partially redundant. Redundancy is complete when we combine Fused and Paired MSAs (they derive from the same single-chain MSAs). Subsequently, the

improved performance is not only due to the larger number of aligned sequences but also to the way those are combined. We try to make this clearer in the discussion.

To further analyse the relationship between the SR and MSA quality, we analyse the relationship between the DockQ score and the interface PPV. The PPV is the number of correct interface contacts divided by the total number of interface contacts, calculated using GaussDCA on the paired alignments and taking the top N DCA signal as contact positions. E.g. If there are 100 true interface contacts, we take the top 100 interface DCA signal positions and calculate how many of these are accurate (see methods).

We find that the SR increases with PPV in the interface using both paired and AF2 MSAs and only paired MSAs. At higher PPVs, the SR seems driven entirely by the PPV, while using both AF2 and paired MSAs outperforms using only the paired MSAs at lower PPVs. The relationship with the interface PPV underlines the importance of pairing the MSA correctly, as the SR is 1 at $PPV > 0.35$ for both MSA modeling strategies.

- As already mentioned above, the comparison of the AF2 performance with that of a single docking algorithm (GRAMM), which moreover doesn't represent the state of the art, is unfair. It may indeed be the case that a fairer comparison may prove the authors right, but this needs to be based on a valid evidence.

We have added MDOCKPP to properly reflect a state-of-the-art comparison. MDOCKPP was ranked first in the server predictions during CASP14-CAPRI for modeling protein complexes.

-The template-based docking procedure used in this study seems to be quite different from the procedure the authors refer to (ref 7). Here, it seems, target complexes from the test

dataset are structurally aligned either to the backbone of the full template complex, or only to the template interface residues. Or in blind predictions, where the structure of the target complex is unknown, template-based 'docking' involves aligning the sequences of each protein of the target complex to that of its homolog in the template complex and going on from there.

We are sorry that an improper wording was used here, we superposed monomers extracted from the target to complexes in the library in order to select templates. We have reformulated this accordingly.

-The legend of Figure 3 reads: 'ROC curve as a function of different metrics for the development dataset (first run). But the text refers to Figure 3 as representing the results for the test set. Indeed, the ROC of Figure 3A represents a plot of the TPR versus FPR and requires scoring & ranking predicted structures for both the TP and TN examples, which the text does not describe for the development set. What then is the difference between the ROCs in Figures 3A and 6A ?

Thank you for this remark. This was an error on our part and we have now made it clear that what is now figure 2 refers to the test set only.

The figures you are referring to, now 2 and 5, refer to different comparisons. The first (now figure 2) refers to separating acceptable models in terms of DockQ score and the second (now figure 5) refers to separating interacting from non-interacting proteins. We have made this more clear in the figure legends and text, e.g.

Figure 5. A) The ROC curve as a function of different metrics for discriminating between interacting and non-interacting proteins.

REVIEWERS' COMMENTS

Reviewer #1 (Remarks to the Author):

No further comments, my previous points have been properly addressed and the manuscript changed to reflect the changes.

Reviewer #2 (Remarks to the Author):

The revised version is largely improved.

The comparison with the performance of another performant docking method MDockPP, adds credibility to the study.

Particularly welcome is the development of the continuous model quality assessment criterion pDockQ, shown here to effectively outperform several versions of the AF2 pLDDT reliability criterion and other simple measures for ranking models and segregating interacting from non-interacting protein pairs.

Also interesting is the deeper analysis of the MSA features contributing to successful prediction, which suggests that interface evolutionary signals as measured by the fraction of interface contacts recalled by DCA, have a strong impact on the prediction results.

Evaluating the performance of AlphaFold-Multimer is a marginal addition given that the corresponding DL model was in fact trained on the dataset used here as the test set.

I have only a few outstanding comments

Results section:

-Development versus test set performance: The significant discrepancy between the performance of AF2 on development set (33.3% - 39.4%) and the test set (57.8% - 58.4%), is striking. This suggests that AF2 performance is dataset dependent. While this is not unexpected, it begs for a comment. It doesn't seem to result from an organism bias, since the AF2 performance on the smaller development dataset is much lower even though it features a higher proportion of bacterial proteins, for which the authors observe a higher SR level in AF2 predictions.

Lines 95-97, 139-140

...'protocol performs quite close to (63% vs 72%) the recently developed AF-Multimer which was developed using the same data as the test set here, making a direct comparison difficult.

'was developed' should be replaced by 'was trained'.

Lines :132-134

The sentence is misleading since the performance of the 3 docking methods is clearly not good.

Suggestion:

Replace "The reason for GRAMM's, TMdock's and MDockPP's good performance is likely due" by "The reason GRAMM's, TMdock's and MDockPP's reach this performance level is likely due"

Lines 362-364

Projecting the fraction of human heterodimers predicted at the current 1% error rate, on the basis of the number of pairwise human PPI in the String DB (11.9 million) is overdoing it, since it is well known that a sizable fraction of the interactions in String are non-physical. The paragraph should be rephrased accordingly.

REVIEWERS' COMMENTS

Reviewer #1 (Remarks to the Author):

No further comments, my previous points have been properly addressed and the manuscript changed to reflect the changes.

We are delighted our changes were satisfactory and thank you for your excellent comments throughout the revision process.

Reviewer #2 (Remarks to the Author):

The revised version is largely improved.

The comparison with the performance of another performant docking method MDockPP, adds credibility to the study.

We are happy this addition was satisfactory and thank you for this suggestion.

Particularly welcome is the development of the continuous model quality assessment criterion pDockQ, shown here to effectively outperform several versions of the AF2 pLDDT

reliability criterion and other simple measures for ranking models and segregating interacting from non-interacting protein pairs.

We thank you for this kind acknowledgement and agree this score is useful for both structure quality assessment and separating interacting from non-interacting proteins.

Also interesting is the deeper analysis of the MSA features contributing to successful prediction, which suggests that interface evolutionary signals as measured by the fraction of interface contacts recalled by DCA, have a strong impact on the prediction results.

Indeed it appears so, that the evolutionary signal in the interface - here measured using DCA - has a strong impact on the outcome, opening up for future improvements in docking by improving this signal.

Evaluating the performance of AlphaFold-Multimer is a marginal addition given that the corresponding DL model was in fact trained on the dataset used here as the test set.

This is correct and we are in full agreement.

I have only a few outstanding comments

Results section:

-Development versus test set performance: The significant discrepancy between the performance of AF2 on development set (33.3% - 39.4%) and the test set (57.8% - 58,4%), is striking. This suggests that AF2 performance is dataset dependent. While this is not unexpected, it begs for a comment. It doesn't seem to result from an organism bias, since the AF2 performance on the smaller development dataset is much lower even though it features a higher proportion of bacterial proteins, for which the authors observe a higher SR level in AF2 predictions.

We thank you for this comment. We agree there is a big difference and have added a comment about this under limitations, where we suggest that performance should be assessed on as large non-redundant datasets as possible to ensure any selection bias does not impact the results. We do not know exactly the origin of the difference. We tried to examine the most obvious differences between the two sets (protein size, species, size of MSAs etc) but did not find anything obvious that separated the sets. Unfortunately, trying to pinpoint the origin of the difference is beyond the goals of this study.

Lines 95-97, 139-140

...'protocol performs quite close to (63% vs 72%) the recently developed AF-Multimer which was developed using the same data as the test set here, making a direct comparison difficult.

'was developed' should be replaced by 'was trained'.'

We have changed this phrasing as suggested to clarify that it was indeed trained.

Lines :132-134

The sentence is misleading since the performance of the 3 docking methods is clearly not good.

Suggestion:

Replace “The reason for GRAMM’s, TMdock’s and MDockPP’s good performance is likely due” by “The reason GRAMM’s, TMdock’s and MDockPP’s reach this performance level is likely due”

We thank you for this suggestion and have changed the phrasing as suggested.

Lines 362-364

Projecting the fraction of human heterodimers predicted at the current 1% error rate, on the basis of the number of pairwise human PPI in the String DB (11.9 million) is overdoing it, since it is well known that a sizable fraction of the interactions in String are non-physical. The paragraph should be rephrased accordingly.

We have changed this statement to reflect a more realistic modeling scenario, which we have also applied in practice (<https://www.biorxiv.org/content/10.1101/2021.11.08.467664v1>).